# Efficient and flexible Integration of variant characteristics in rare variant association studies using integrated nested Laplace approximation

**Hana Susak**[1,2,3], **Laura Serra-Saurina**[4,5,6], **German Demidov**[1,7,8], **Raquel Rabionet**[1,9], **Laura Domènech**[1,4], **Mattia Bosio**[1], **Francesc Muyas**[1,7,8], **Xavier Estivill**[1,10], **Geòrgia Escaramís**[1,4,11]☯*, **Stephan Ossowski**[1,7,8]☯

**1** Centre for Genomic Regulation (CRG), The Barcelona Institute of Science and Technology, Barcelona, Spain, **2** Division of Computational Genomics and Systems Genetics, German Cancer Research Center (DKFZ), Heidelberg, Germany, **3** European Molecular Biology Laboratory, Genome Biology Unit, Heidelberg, Germany, **4** Biomedical Research Networking Centre consortium of Public Health and Epidemiology (CIBERESP), Madrid, Spain, **5** Center for research in occupational Health (CiSAL), Department of Experimental and Health Sciences, Universitat Pompeu Fabra, Barcelona, Spain, **6** Research Group on Statistics, Econometrics and Health (GRECS), Universitat de Girona (UdG), Girona, Spain, **7** Universitat Pompeu Fabra (UPF), Barcelona, Spain, **8** Institute of Medical Genetics and Applied Genomics, University of Tübingen, Tübingen, Germany, **9** Department of Genetics, Microbiology and Statistics, Faculty of Biology, IBUB, Universitat de Barcelona; CIBERER, IRSJD, Barcelona, Spain, **10** Women's Health Dexeus, Barcelona, Spain, **11** Departament de Biomedicina, Facultat de Medicina i Ciències de la Salut, Institut de Neurociències, Universitat de Barcelona, Spain

☯ These authors contributed equally to this work.
* gescaramis@ub.edu

**Data Availability Statement:** The authors confirm that all non access-restricted data underlying the findings are fully available without restriction. 1000

## Abstract

Rare variants are thought to play an important role in the etiology of complex diseases and may explain a significant fraction of the missing heritability in genetic disease studies. Next-generation sequencing facilitates the association of rare variants in coding or regulatory regions with complex diseases in large cohorts at genome-wide scale. However, rare variant association studies (RVAS) still lack power when cohorts are small to medium-sized and if genetic variation explains a small fraction of phenotypic variance. Here we present a novel Bayesian rare variant Association Test using Integrated Nested Laplace Approximation (BATI). Unlike existing RVAS tests, BATI allows integration of individual or variant-specific features as covariates, while efficiently performing inference based on full model estimation. We demonstrate that BATI outperforms established RVAS methods on realistic, semi-synthetic whole-exome sequencing cohorts, especially when using meaningful biological context, such as functional annotation. We show that BATI achieves power above 70% in scenarios in which competing tests fail to identify risk genes, e.g. when risk variants in sum explain less than 0.5% of phenotypic variance. We have integrated BATI, together with five existing RVAS tests in the 'Rare Variant Genome Wide Association Study' (rvGWAS) framework for data analyzed by whole-exome or whole genome sequencing. rvGWAS supports rare variant association for genes or any other biological unit such as promoters, while allowing the analysis of essential functionalities like quality control or filtering. Applying rvGWAS

Genome Project vcf data is available from the ftp://ftp.1000genomes.ebi.ac.uk/vol1/ftp/release/20130502/ server. Sequencing data and variants of CLL individuals have been deposited at the European Genome-Phenome Archive (EGA, http://www.ebi.ac.uk/ega/) under accession numbers EGAS00000000092 and EGAS00001003027. Access-restricted data from patient samples included in the in-house dataset (germline variants for Spanish cohort) have been deposited at the European Genome-Phenome Archive EGA under the accession numbers EGAD00001006371, EGAD0000106370, EGAD00001005288, and EGAD00001004808. Additional patient data can be made available upon request to the Data Protection Officer at the Fundació Centre de Regulació Genòmica (dpo@crg.eu). Software is available from https://github.com/hanasusak/rvGWAS.

**Funding:** SO, HS, XE, FM and GD received funding from the European Union's Horizon 2020 research and innovation programme under grant agreement No 635290 (PanCanRisk). SO and GD received funding from the European Union's Horizon 2020 research and innovation programme under grant agreement No 779257 (Solve-RD). RR received support from the Fundació La Marató 70/307:201726. HS, GD, RR, LD, MB, FM, XE, GE and SO received support of the Spanish Ministry of Economy and Competitiveness, 'Centro de Excelencia Severo Ochoa 2013-2017, and the CERCA Programme / Generalitat de Catalunya. The funders had no role in study design, data collection and analysis, decision to publish, or preparation of the manuscript.

**Competing interests:** The authors have declared that no competing interests exist.

to a Chronic Lymphocytic Leukemia study we identified eight candidate predisposition genes, including EHMT2 and COPS7A.

## Author summary

Complex diseases are characterized by being related to genetic factors and environmental factors such as air pollution, diet etc. that together define the susceptibility of each individual to develop a given disease. Much effort has been applied to advance the knowledge of the genetic bases of such diseases, specially in the discovery of frequent genetic variants in the population increasing disease risk. However, these variants usually explain a little part of the etiology of such diseases. Previous studies have shown that rare variants, i.e. variants present in less than 1% of the population, may explain the rest of the variability related to genetic aspects of the disease. Genome sequencing offers the opportunity to discover rare variants, but powerful statistical methods are needed to discriminate those variants that induce susceptibility to the disease. Here we have developed a powerful and flexible statistical approach for the detection of rare variants associated with a disease and we have integrated it into a computer tool that is easy and intuitive for the researchers and clinicians to use. We have shown that our approach outperformed other common statistical methods specially in a situation where these variants explain just a small part of the disease. The discovery of these rare variants will contribute to the knowledge of the molecular mechanism of complex diseases.

## Introduction

The rapidly improving yield and cost-effect ratio of Next Generation Sequencing (NGS) technologies provide the opportunity to study associations of genetic variants with complex multi-factorial diseases in large cohorts at a genome-wide scale. As opposed to genome-wide association studies (GWAS), which are based on counting of genotypes at predefined genomic positions with alternative alleles of medium to high minor allele frequency in the population (MAF >1%), whole-exome and whole-genome sequencing (WES, WGS) enable the study of rare genetic variants (RV) across the whole exome or genome, respectively. Previous studies have shown that RVs play an important role in the etiology of complex genetic diseases[1–4]. Furthermore, it has been demonstrated that RVs are more likely to affect the structure, stability or function of proteins than common variants[5,6]. Therefore, statistical analysis of the combined set of rare variants across genes or regulatory elements has the potential to reveal new insights into the genetic heritability of complex diseases and the predisposition to cancer. To this end, rare variant association studies (RVAS) that facilitate identification of novel disease loci based on the burden of rare and damaging variants with low to medium effect size within genomic units of interest have been developed [7].

One of the major difficulties when associating rare variants to disease is the lack of power when using traditional statistical methods like GWAS. Given that few individuals are carriers of the rare alternative allele, association studies based on single variant positions would require extremely large sample sizes. To overcome this obstacle and to increase statistical power, studies of RV consider simultaneously multiple variable positions within functional biological units, such as genes, promoters or pathways, for association to disease. Different statistical methods that address the problem of aggregated analysis of rare variants in case-control

studies have been proposed. For example, score based methods pool minor alleles per unit into a measure of burden, which is used for association with a disease or phenotypic trait[8–11]. These burden tests are powerful when a high proportion of RVs found in a gene affect its function and their effects on the disease are one-sided, i.e. either protective or deleterious. This is rarely the case since usually few deleterious variants coexist with many neutral and possibly some protective variants. Hence advanced methods have been developed to consider heterogeneous effects among RVs on the disease (or trait), which are mainly based on variance component tests, e.g. SKAT and C-alpha[12,13]. These methods are more powerful than burden tests when the assumption of unidirectional effects does not hold[14]. More recently, novel methods have been introduced. These allow that both types of genetic architectures may coexist throughout the genome, by being constructed as a linear combination between burden and variance-component tests, such as SKAT-O[15]. He et al.[16] developed an alternative method, a hierarchical Bayesian multiple regression model (HBMR) additionally accounting for variant detection errors commonly produced using NGS data, by incorporation of genotype misclassification probabilities in the model. Sun et al.[17] proposed a mixed effects test (MiST) within the framework of a hierarchical model, considering biological characteristics of the variants in the statistical model. In brief, MiST assumes that individual variants are independently distributed, with the mean modeled as a function of variant characteristics and random effects that account for heterogeneous variant effects on phenotype caused by unknown factors. In the resulting generalized linear mixed effects model (GLMM) variant-specific effects are treated as the random part of the model and patient and variant characteristics as the fixed part. The authors claim that, under the assumption that associated variants share common characteristics such as similar impact on protein function (e.g. primarily loss of function), using this prior information increases the power of the test. However, they also note that attempting to estimate the full model for inference purposes requires multiple integration, such that it becomes too computationally intensive for a genome-wide scan. Instead, a score test under the null hypothesis of no association is proposed, avoiding multiple integration.

Building on the concept of MiST, but with the motivation of making inference based on full model estimation, we propose a Bayesian alternative to the GLMM, using the Integrated Nested Laplace Approximation (INLA) for efficient model estimation[18]. Calculating the marginal likelihood to estimate complex models in a fully Bayesian manner is often infeasible. Therefore, approximate procedures such as the heuristic Markov Chain Monte Carlo (MCMC) method are conventionally applied[16]. MCMC is a highly flexible approach that can be used to make inference for any Bayesian model. However, evaluating the convergence of MCMC sampling chains is not straightforward[19]. Another concern with MCMC is the extensive computation time, especially in large-scale analyses such as genome-wide scans. INLA is a non-sampling based numerical approximation procedure, developed to estimate hierarchical latent Gaussian Markov random field models. Being based on numerical approaches instead of simulations renders INLA substantially faster than MCMC. Furthermore, Rue and Martino[20] demonstrated for several models that INLA is also more accurate than MCMC when given the same computational resources. The flexibility of modeling within the Bayesian framework combined with rapid inference approaches opens new possibilities for genetic association testing.

Here, we present a novel Bayesian rare variant Association Test using INLA (BATI), implemented as part of the 'Rare Variant Genome Wide Association Study' (rvGWAS) framework. rvGWAS combines quality control (QC), interactive filtering, detection of data stratification (technical or population based), integration of functional variant annotations and four commonly used rare variant association tests (Burden, SKAT-O, KBAC and MiST) as well as the two Bayesian alternatives, HBMR and BATI. We demonstrate using realistic benchmarks that BATI substantially outperforms existing methods if prior information on the effect of variants

on protein function is used. We further show that BATI successfully copes with complex population structure and other confounders. Finally, we propose how to use 'difference in deviance information criterion' (ΔDIC) for model selection.

## Material and methods

### Bayesian rare variant Association Test based on Integrated nested Laplace approximation (BATI)

Integrated Nested Laplace Approximation is a recent approach to implement Bayesian inference on latent Gaussian models, which are a versatile and flexible class of models ranging from (generalized) linear mixed models (GLMMs) to spatial and spatio-temporal models. A detailed definition of INLA can be found in[18,21,22]. Here we applied INLA using the implementation of the R-INLA project (R package INLA version 17.06.20) to build a hierarchical Bayesian approach to the GLMM for the association of rare variants with phenotypes in the context of case-control studies. Our method termed BATI can efficiently and flexibly integrate a large number of categorical and numeric characteristics of genetic variants as covariates, as INLA facilitates estimation of the full model even for complex structures of random effects.

### Model specification

Assume we have N individuals, and let $Y_i$ $(i = 1,\ldots,N)$ be the observed phenotype of the $i_{th}$ individual that belongs to an exponential family:

$$Y_i \sim \pi(Y_i; \mu_i, \theta) \tag{1}$$

where the expected value $\mu = E(Y_i)$ is linked to a linear predictor $\eta_i$ through a known link function $g(\cdot)$, so that $g(\cdot) = \eta_i$. In our case $Y_i$ is a binary variable representing affected individuals (cases) vs. unaffected individuals (controls). We propose to construct the likelihood of the data based on a logistic distribution and use the identity function for $g(\cdot)$. The linear predictor $\eta_i$ is defined to account for potential confounding covariates at the individual level as well as for covariates at the variant level such as a variant's functional impact:

$$\eta_i = X_i^t \alpha + G_i^t \beta \tag{2}$$

where $X_i$ is a $m \times 1$ vector of individual-based confounding covariates, such as gender, age or ethnicity, and $G_i$ denotes a $p \times 1$ vector of genotypes for $p$ RVs. Each genotype is coded as 0, 1, or 2, representing the number of minor alleles. $\alpha$ is the regression vector of coefficients that represent the effects of an individual's covariates on phenotype and $\beta$ is the regression vector of coefficients reflecting the genetic variant's effect on phenotype.

BATI can account for individual variant characteristics under the assumption that similar variant-specific characteristics, such as similar functional impact scores or gene annotation categories (missense, LoF, splice-donor/acceptor, InDel, regulatory), have a similar effect on the function of the protein and hence the phenotype, while still allowing for potential variant-specific heterogeneity effects. Thus $\beta$ can be modeled in a hierarchical way as:

$$\beta_j = Z_j^t \omega + \delta_j \tag{3}$$

where $Z^t$ is a $p \times q$ matrix (for $q$ different variant characteristics, i.e. each row of this matrix represents a specific functional annotation of a single variant), $\omega$ is a vector of $q \times 1$ $(j = 1,\ldots,q)$ variant-specific regression coefficients leveraging variant effects on phenotype based on variant characteristics, and $\delta$ is a $p \times 1$ random effects vector representing unknown factors leading to heterogeneous variant effects on phenotype. The random effects vector is assumed to follow a

multivariate Gaussian distribution with mean 0 and covariance matrix $\tau Q$. If no dependency structure is defined across variants, $Q$ is a $p \times p$ identity matrix and $\tau$ the random effects variance. Even though correlation structure across variants is not considered in this work, it is worth mentioning that INLA has the potential to estimate the elements of $Q$ in such a way that it would reflect the dependency structure. This is enabled by INLA because it provides Laplace approximation of the posterior distributions, potentially enabling the estimation of the full model for complex structures of random effects.

Plugging Eq (3) into (2) we obtain the expression of a generalized linear mixed effects model (GLMM):

$$\eta_i = X_i^t \alpha + (G_i^t Z)\omega + G_i^t \delta \tag{4}$$

with $\alpha$ and $\omega$ as fixed effects coefficients and $\delta$ as random effects coefficients. Given the vector of parameters $\theta = \{\alpha, \omega, \delta\}$, the objectives of the Bayesian computation are the marginal posterior distributions for each of the elements of the parameter vector $p(\theta_s | y)$ and for the hyper-parameter $p(\tau | y)$. In order to compute the marginal posterior for the parameters, we first need to compute $p(\tau | y)$ and $p(\theta_s | \tau, y)$. The INLA approach exploits the assumptions of the model to produce a numerical approximation to the posteriors of interest, based on the Laplace approximation[23].

## Model selection

The classical approaches of association tests are based on hypothesis testing, where the null hypothesis assumes no genetic effects, and the alternative hypothesis assumes a genetic effect on the phenotype. In the context of BATI this can be specified as follows:

$$H_0 : \eta_i = X_i^t \alpha \tag{5}$$

$$H_1 : \eta_i = X_i^t \alpha + (G_i^t Z)\omega + G_i^t \delta \tag{6}$$

A classic Bayesian criterion for model goodness of fit is the *Deviance Information Criterion* (*DIC*)[24]. *DIC* is calculated as the expectation of the deviance over the posterior distribution plus the effective number of parameters. Thus, difference in *DIC* between the $H_0$ and the $H_1$ models, $\Delta DIC = DIC_{H_0} - DIC_{H_1}$, can be used as the model selection criterion. As a rule of thumb values of $\Delta DIC > 10$ are recommended to reject the null-hypothesis. However, to evaluate the ability of $\Delta DIC$ to correctly choose between null or alternative models we suggest the use of simulations, as proposed by Holand et al.[25]. To find an estimate of the probability of type I error, concluding that there are genetic effects when in truth there is none, we randomly assign individuals to either cases or controls. We then adjust models under the null and the alternative hypothesis for each gene or biological unit included in the genome wide study, obtaining the empirical distribution of $\Delta DIC$. Finally, we rank the genes by $\Delta DIC$ values in ascending order and select a $\Delta DIC$ threshold from the quantile corresponding to the desired significance level. For instance, if a significance level of 0.1% is desired, we pick the $\Delta DIC$ value of the gene ranked at top 0.1% position as the significance threshold. For more robust threshold estimation, we propose to generate $S$ datasets by randomly shuffling cases and controls, such that $S \Delta DIC$ thresholds can be obtained and the median of the thresholds can be used. We used $S = 10$ for model selection in our benchmark study.

## A comprehensive framework for rare variant association analysis (RVAS)

We developed the 'Rare Variant Genome Wide Association Study' (rvGWAS) framework (Fig 1A and S1 Fig), an all-in-one tool designed for RVAS tests using case-control cohorts analyzed

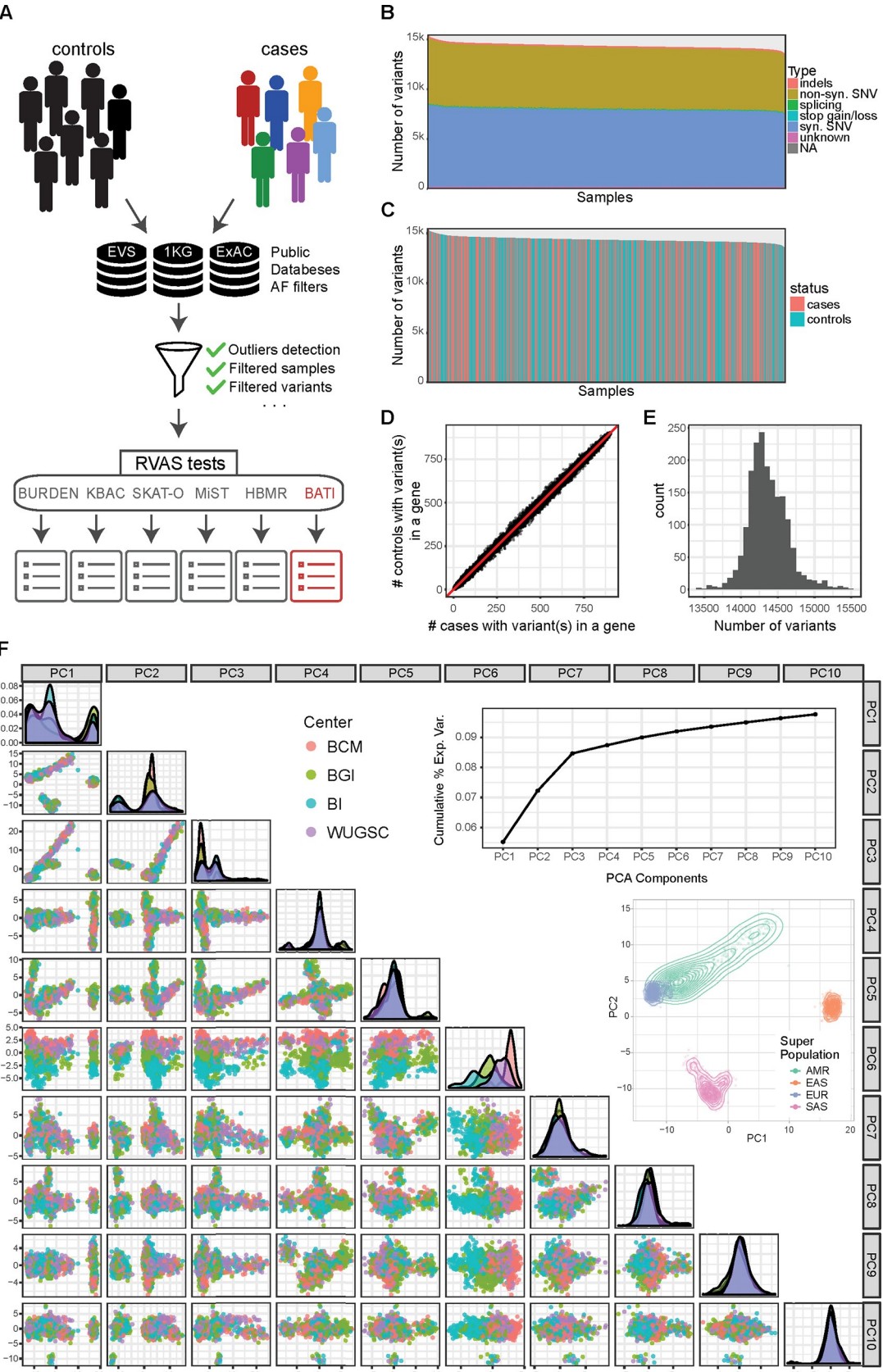

**Fig 1. rvGWAS workflow and QC plots for 1810 high quality samples from 1000GP used for benchmarking.** The rvGWAS workflow is exemplary shown for a simulated breast cancer case-control cohort based on 1810 whole exome sequencing

 

datasets from 1000GP. (**A**) rvGWAS workflow for performing QC and six RVAS tests. Different colors indicate that cases carry different potentially damaging rare variants within the same functional biological unit, such as a gene. The QC module computes quality statistics shown in panels B-F. The result of each RVAS test is a ranked list of genes with various informative attributes. (**B**) Bar-plot for number of variants per sample, colored by functional annotation of variants. (**C**) Barplot for number of variants per sample, colored by assignment to cases (~1/2) or controls (~1/2). (**D**) Number of variants per gene in cases (x-axis) and controls (y-axis). Each dot is one gene, while the red line shows the ratio of the number of cases and controls (1:1). (**E**) Histogram for number of mutations per sample after removal of outliers. (**F**) Projection on first 10 PCA components. Samples are colored by sequencing center. The graph in the upper right corner shows the cumulative percentage of variance explained per principal component. Principal components can be used as covariates in several RVAS tests.

by NGS. rvGWAS supports rare variant association aggregating by genes or any other biological unit such as promoters or enhancers. It provides all essential steps and functionalities to perform the complete analysis of whole-exome sequencing (WES) or whole-genome sequencing (WGS) based case-control study designs: (1) it facilitates comprehensive quality control and filtering, (2) it evaluates data stratification (either technical or population based), (3) it enables the integration of patient- and/or variant-based covariates in association tests in an easy and intuitive fashion, and (4) it integrates six conceptually different rare-variant association methods. It is implemented in a modular way and provides great flexibility, allowing for the analysis of a wide range of association study designs.

BATI and five other RVAS methods are integrated in the rvGWAS framework. KBAC, SKAT-O, and MiST, were chosen to be included due to their superior performance compared to eight other RVAS methods in a benchmark study by Moutsianas et al.[14]. In addition, we included the classical Burden test representing the most simple and intuitive form of RVAS tests. Finally, we incorporated HBMR, which is conceptually the most similar to BATI in terms of its estimation approach (while MiST is more similar in terms of model specification). The six supported RVAS tests represent a broad spectrum of approaches, including classic aggregation of variants as a Burden variable, variance component bidirectional tests, mixed effect models and Bayesian inference.

rvGWAS is implemented as a pipeline of R scripts, and is available online at https://github.com/hanasusak/rvGWAS. Detailed descriptions of the tool, included methods as well as parameters are provided in S1 Text.

## 'Semi-synthetic' simulations of whole-exome sequencing based case-control studies

To allow for benchmarking using highly realistic disease cohorts, which correctly represent all expected sources of noise, we developed a new disease cohort simulator combining thousands of real WES datasets from various studies with known risk variants for a selected disease type. The simulator randomly assigns WES samples to the case or control group and introduces predisposition variants found in ClinVar or the Human Gene Mutation Database (HGMD) [26] for a disease of choice into the VCF files of cases.

We used two large datasets as basis for the simulation: 1) WES data of the 1000 Genomes Project (1000GP), and 2) an in-house dataset combining patients diagnosed with various conditions and healthy individuals subjected to WES during 2012 to 2017. VCF files from 1000GP (phase3)[27,28] were downloaded from ftp://ftp.1000genomes.ebi.ac.uk/vol1/ftp/release/20130502/. This cohort contains 2504 individuals from 26 populations. WES libraries of 1000GP were prepared using one of four oligo enrichment kits: (1) Nimblegen SeqEz V2, (2) Nimblegen SeqEz V3, (3) VC Rome, and (4) Agilent SureSelect V2. Additional sample information used as covariates (population, super population, gender) was obtained from the file integrated_call_samples_v3.20130502.ALL.panel. We excluded related individuals, e.g in

parent-child trios we included the parents (if not consanguineous), but not the child. To minimize issues with population stratification due to highly diverse populations we only included individuals not belonging to African ancestry populations, as Africans had on average 25% more variants than individuals from other ancestry groups. Nonetheless, the remaining cohort still represents a mixed population, allowing us to benchmark the RVAS tests' performances on genetically diverse populations.

The in-house 'Iberian' WES cohort includes 1189 individuals of Spanish ancestry and is highly homogeneous as demonstrated by PCA (S2F Fig). WES libraries were prepared using three different oligo enrichment kits: (1) Agilent SureSelect 50, (2) Agilent SureSelect 71, and (3) Nimblegen SeqEz V3. Computational analysis and variant calling was performed according to GATK best practice guidelines (https://software.broadinstitute.org/gatk/best-practices/)). For simulation purposes we only considered genomic loci that were targeted and covered with at least 10 sequence reads by all oligo enrichment kits, and variants with a call rate higher than 85%. Samples that were identified as outliers based on the number of called variants, transition to transversion (Ti/Tv) ratio, or their projection on the first two principal components from principal component analysis were removed from further analysis. The filtered datasets, named 1000GP and Iberian cohort, consisted of 1,810 and 1,167 samples harboring 493,314 and 285,658 unique loci with alternative alleles, respectively. From 1000GP we randomly selected half of the samples as cases, the other half as controls. An important technique for adding power to case-control studies, especially when the number of cases is limited, is to enroll more than one control per case, although it has been shown that little is gained by including more than two controls per case[29]. Therefore, we chose to use one case per two controls in the relatively small Iberian cohort.

## Simulating a breast cancer risk cohort

To introduce realistic disease variants into a 'semi-synthetic' breast cancer predisposition cohort, we queried the ClinVar and HGMD databases for breast cancer risk variants annotated as exonic or splicing. We removed variants that had MAF higher than 0.01 in any ancestry population in any of three commonly used exome databases: EVS, 1000GP or ExAC. Six genes had more than five annotated disease risk variants in ClinVar and HGMD: *BRCA2 (MIM: *600185), BRCA1 (MIM: *113705), PALB2 (MIM: *610355), BRIP1 (MIM: *605882), CHEK2 (MIM: +604373)* and *BARD1 (MIM: *601593)* (S1 Table), which we used as a pool of variants to simulate risk patients by adding variants to the VCF files (zero or one variant per case). As expected, all six genes already had rare variants, likely benign, in the unmodified cohorts (S2 and S3 Tables). This type of noise is expected in any case-control study using WES data, and hence makes the simulation more realistic.

Using the 1000GP cohort we simulated a rare variant etiology by ethnicity scenario. Samples in assigned cases and controls were separated into populations according to 1000GP annotation and each variant from the pool was assigned to one of the super populations; EUR +AMR, SAS or EAS. Variants annotated in ExAC were assigned to the population with the largest observed allele frequency and variants not observed in ExAC were assigned to one of the three populations randomly. In the case of the Iberian cohort, as we have an homogeneous population, variants were assigned without any stratification.

We generated three genetic architectures per gene, with ~2% (1), ~1% (2) or ~0.5% (3) of phenotypic variance explained (VE) by introducing ClinVar and HGMD risk variants. To this end we used the method of So et al.[30] for calculation of cumulative VE each time a variant was added to a gene until the targeted VE was reached. Calculation of VE requires three parameters per each variant: the prevalence of the trait, the population frequency of the risk

allele, and the genotype relative risk (RR). In practice, only odds ratios (OR) are available in many case-control studies. However, OR approximates RR when the disease prevalence in a population is low[30]. We selected an estimate for the breast cancer prevalence 5 years' period of adult Spanish women (0.66%), defined as the percentage of current cases (new and preexisting) over the specified period of time. This estimate is obtained from Global Cancer Observatory website https://gco.iarc.fr/today/online-analysis-table [31]. In order to generate realistic RR distributions, we generated a distribution (S3 Fig) assuming that the likelihood of having high RR is negatively correlated with MAF[14]. For *BRCA1* and *BRCA2* we simulated two different types of genetic architectures, by introducing in one architecture only missense variants, and in the other only loss of function (LoF) SNVs (i.e. stop-gain, stop-loss or splicing). This allowed us to test if MiST and BATI benefit from features that capture biological function and context of variants. For the four remaining genes, the variants were simulated regardless of their functionality. The simulation procedure is repeated 100 times for each of the 8 architectures in order to generate 100 datasets for evaluation of statistical power and type I error rates (TIER).

## Results

### Quality control and filtering of benchmark WES cohorts

Cohorts used for benchmarking of test methods consisted of 1,810 individuals in the 1000GP cohort and 1,167 individuals in the Iberian cohort, harboring 493,314 and 285,658 unique loci, respectively, deviating from the GRCh37 (hg19) reference genome, but only including SNVs and short InDels. Both datasets were analyzed and filtered using the rvGWAS quality control modules (see Material and Methods and S1 Text). For benchmarking purposes, we only considered variants in regions targeted by all used oligo enrichment kits. However, in the case of the Iberian cohort we observed that a small subset of regions supposed to be targeted consistently showed low coverage in a kit-specific manner, leading to strong biases identified by the data stratification module of rvGWAS. The bias disappeared when excluding regions with less than 10x average coverage in at least one kit (S2F Fig). Samples included in the final simulation cohorts show no biases in any of the first ten components of the PCA (1000GP: Fig 1F, Iberian: S2F Fig), and the explained variance per PCA component is low (Figs 1F and S2C). Furthermore, samples in the two cohorts show a normal distribution of the number of mutations (Figs 1E and S2E) and show no bias in the number of variants and fractions of InDels or synonymous, nonsynonymous and LoF SNVs (Figs 1B,1C, S2A and S2B). Finally, there is a high correlation between the fraction of cases and of controls having variants in any given gene (Figs 1D and S2D).

### Benchmarking RVAS Tests using semi-synthetic breast cancer risk cohorts

We used the rvGWAS framework to benchmark the six RVAS tests (Burden, SKAT-O, KBAC, MiST, HBMR and BATI) on the 1000GP and Iberian cohorts with simulated breast cancer risk variants. In order to simulate a realistic breast cancer predisposition case-control study we randomly split each of the original cohorts in a case (1000GP: 905, Iberian: 389 samples) and a control group (1000GP: 905, Iberian: 778 samples), and, in the case group samples, added ClinVar and HGMD risk variants to the genes *BRCA2, BRCA1, PALB2, BRIP1, CHEK2* and *BARD1* using realistic variance explained (VE) rates (see Material and Methods). Before performing the RVAS we filtered out common variants (AF>0.01 in public databases or in the randomized control group) as well as variants that were annotated as synonymous or had a CADD score below 10 (likely benign, see https://cadd.gs.washington.edu/info). For BATI and MiST we used prior information on variant characteristics as covariates: CADD scores as a

quantitative variable and exonic function (missense, loss-of-function, InDels) as a categorical variable. We repeated the simulation and benchmarking process 10 times, including the randomized case-control assignment in order to randomize background noise in each benchmark cycle.

## Type I error rate estimates

The six benchmarked RVAS tests use diverse criteria for statistical significance (p-value, Bayes factor or $\Delta DIC$). To generate comparable significance thresholds, we performed RVAS tests on randomly split cohorts, but without introduced ClinVar and HGMD risk variants. Hence, significant associations should only be found by random chance and constitute false positives. This procedure allowed us to obtain comparable thresholds for desired type I error rates for all methods. For each of the 10 random cohort splits we obtained p-value significance thresholds for Burden, KBAC, SKAT-O and MiST that translate to 5%, 0.1% and 0.01% TIER. Similarly, for HBMR and for BATI we calculated thresholds for Bayes factor and $\Delta DIC$ resulting in the same TIER levels. Estimated thresholds are highly similar across all 10 randomized case-control splits (S4 Fig). At 0.01% TIER only 2 genes (out of ~20,000) are expected as significant by chance, therefore the observed small fluctuation of estimated significance thresholds is not surprising. We finally used the test-specific median from 10 random splits as thresholds to label a gene as significant for subsequent power analyses (S4 Fig and Tables 1 and S4).

We noticed that MiST shows zero inflated p-values (S5A Fig). These unexpected zero p-values occur exclusively for genes with few variants (<10) across the cohort, indicating that the MiST method fails to obtain accurate p-values for genes with low burden of variants. Hence, we removed all genes with p-value 0 from MiST results (S5B Fig). No other method showed a p-value inflation artefact or unexpectedly high Bayes Factor or $\Delta DIC$ values (S5C–S5G Fig).

## Power analysis for six RVAS test methods

We next determined the power of the competing RVAS tests to identify the 8 breast cancer risk genes (*BRCA1-Missense*, *BRCA1-LoF*, *BRCA2-Missense*, *BRCA2-LoF*, *PALB2*, *BRIP1*, *CHEK2* and *BARD1*) at the three TIER levels 5%, 0.1% and 0.01% and at three levels of VE of 2%, 1% and 0.5% (1000GP: Fig 2, Iberian: S6 Fig). For the 1000GP cohort we found that all methods showed a power close to 100% at a TIER of 5% across all tested VE levels, except for Burden and SKAT-O, which showed decreased performance for VE = 0.5% (Fig 2A–2C left). Testing 20,000 genes (whole exome) at a TIER of 5% we expect around 1000 false positive genes, which is a poor choice for most studies. Using a TIER of 0.1% (~20 false positive genes expected), differences between the tests become more pronounced, with Burden, KBAC, SKAT-O and MiST showing decreased power already for 1% VE, and all methods showing

**Table 1. P-value, Bayes Factor (HBMR) and ΔDIC (BATI) thresholds for Type I error rates (TIER) of 0.05, 0.001 and 1e-04 estimated on 1000GP.** We randomly permuted case and control labels 10 times and for each estimated empirical thresholds for each RVAS test. The median TIER values from 10 random permutations are used as thresholds for benchmark comparison.

| Method (statistical criteria) | 0.05 TIER | 0.001 TIER | 1e-04 TIER |
|---|---|---|---|
| BURDEN (p-value) | 0.0519 | 1.12e-03 | 7.79e-05 |
| KBAC (p-value) | 0.0650 | 1.52e-03 | 1.52e-04 |
| SKAT-O (p-value) | 0.0563 | 1.47e-03 | 1.66e-04 |
| MiST (p-value) | 0.0766 | 2.26e-09 | 3.33e-16 |
| HBMR (Bayes Factor) | 1.2678 | 3.5774 | 9.0838 |
| BATI (ΔDIC) | 2.3898 | 9.5929 | 14.4623 |

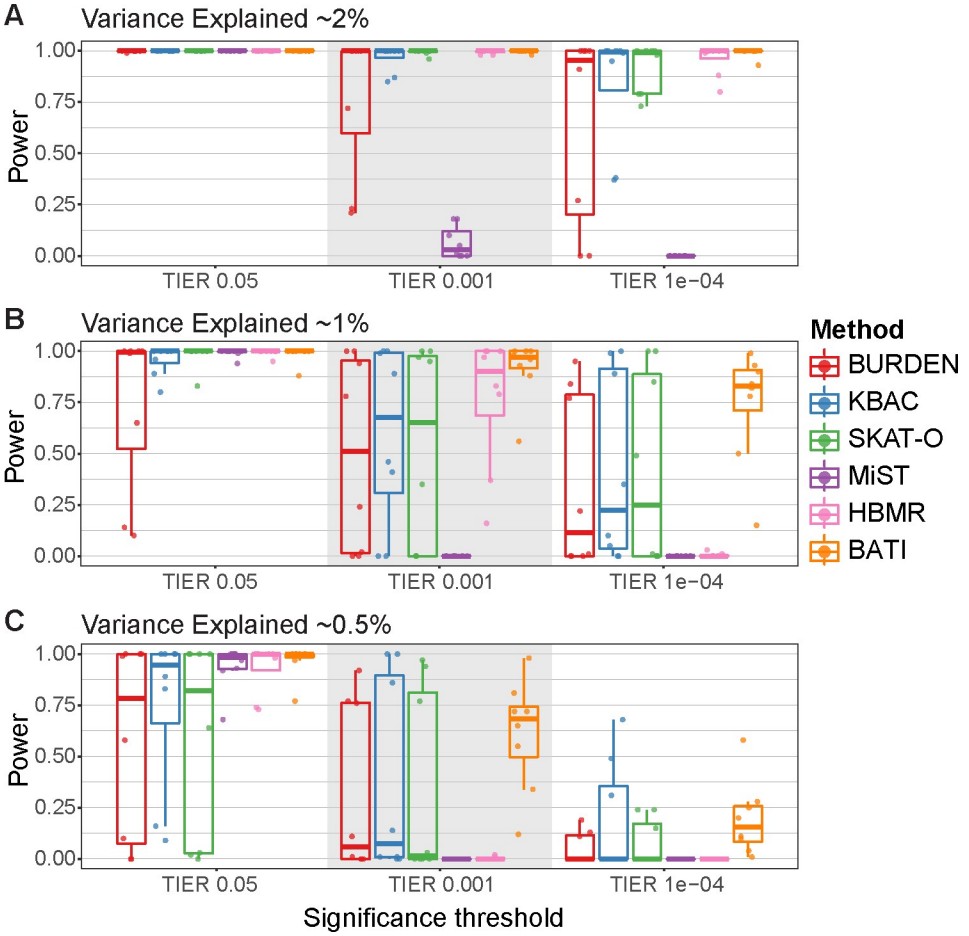

**Fig 2. Benchmarking power of RVAS methods for the 1000GP-based BRCA risk study.** Each dot in the plots represents one of eight simulated risk genes, and y-axis values show the fraction of 100 simulations in which the gene was called as significant. Boxplots represent quantile distributions of significant fraction, where the box depicts the interquartile range of this information. RVAS tests were benchmarked under the following 9 settings. Variance explained (VE) of the incorporated risk variants is (A) ~2%, (B) ~1%, and (C) ~0.5%. For each VE we tested three type I error rate levels (TIER), left: TIER 5%, middle: TIER 0.1%, and right: TIER 0.01%.

decreased power at 0.5% VE (Fig 2A–2C middle), however BATI maintains the highest median power of 70% at TIER 1%. Using a strict TIER of 0.01% (2 false positives expected for the whole exome), all tools except for Burden and MiST are able to identify risk genes at 2% VE at almost 100%. However, performance of all methods except BATI drops substantially for 1% VE. At 0.5% VE most methods miss the majority of risk genes in the majority of simulations (median power close to zero) except for BATI, which maintains a power of 15% (Fig 2A–2C right). Note that MiST performed very poorly for the strict TIER thresholds of 0.1% and 0.01%, likely due to the aforementioned zero-p-value inflation issue, which results in a large number of false positives.

Results are mostly similar in the benchmark using the Iberian cohort (S6 Fig).

## Risk gene-wise power analysis

Each gene has a different architecture, i.e. rate of (likely benign) rare variants in the original cohorts, functional impact estimates for known risk variants, fraction of stop-gain or splicing

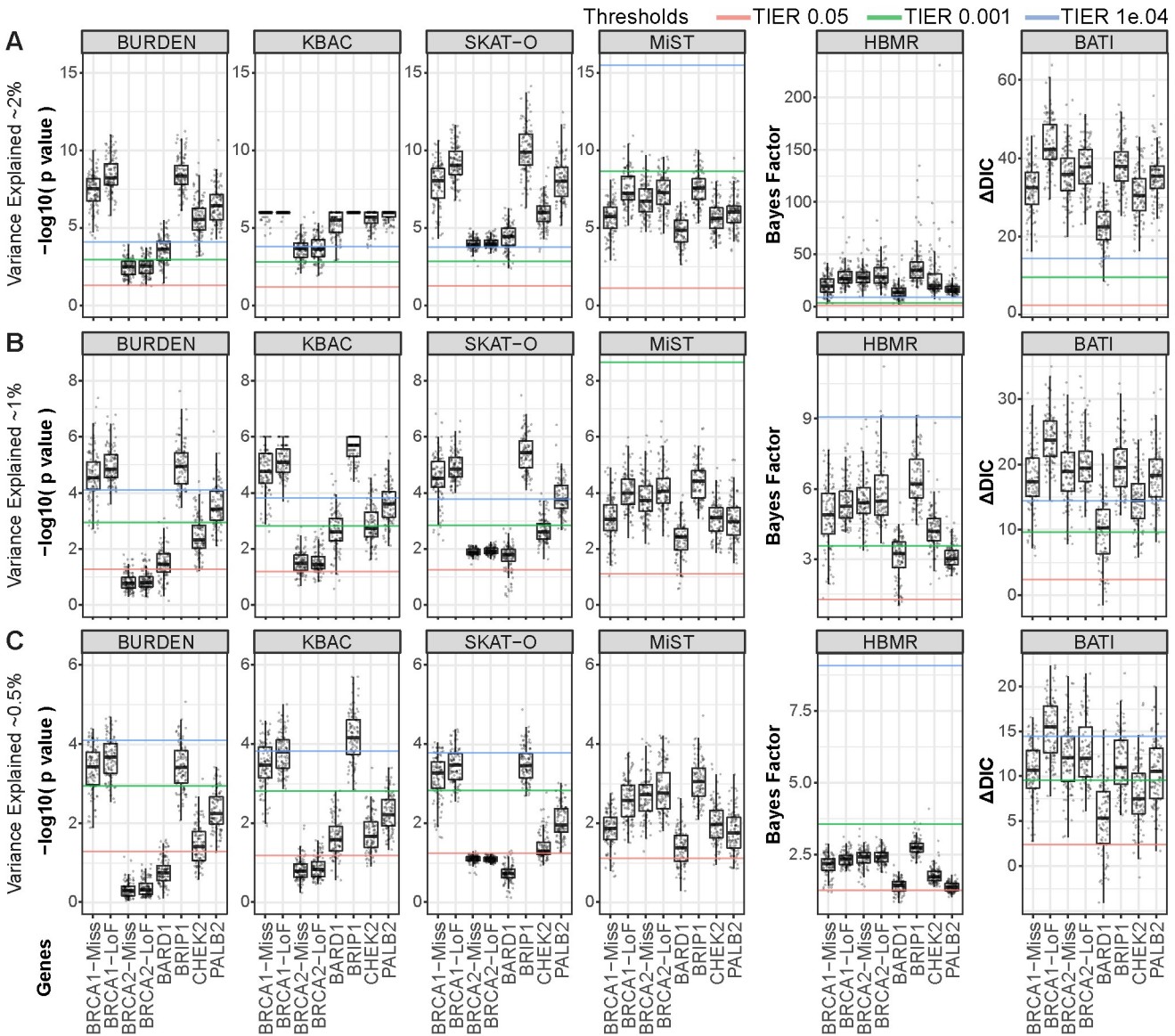

**Fig 3. Benchmarking statistical power to detect rare variant associations for 8 genes individually.** Rare variants annotated for increased breast cancer risk were simulated into the 1000GP dataset following a rare variant etiology by ethnicity scenario. Samples were separated into populations according to 1000GP annotation and each variant from the pool of ClinVar and HGMD variants was assigned to one of the super populations; EUR+AMR, SAS or EAS. Power (y-axis) per gene for 6 methods (Burden, KBAC, SKAT-O, MiST, HBMR and BATI) is shown for (A) 2%, (B) 1%, and (C) 0.5% variance explained between cases and healthy controls. Lower, middle and upper lines indicate relaxed (5%), medium (0.1%) and strict (0.01%) TIER thresholds, respectively.

variants etc. We therefore benchmarked the performance of all RVAS tests across 100 simulations of risk variants for each gene separately (1000GP cohort: Fig 3 and Table 2, Iberian cohort: S7 Fig). In the gene-wise power plots we indicate the three TIER thresholds using red (5%), green (0.1%) and blue (0.01%) lines. Note that due to different y-Axis scaling these lines are not on the same height for different tests. All methods except Burden and MiST identify all risk genes at 0.01% TIER in the 2% VE setting. However, substantial differences in power of the tests appear when VE is only 1% or 0.5%. While BATI calls most genes with TIER 0.01% with >88% power except for BARD1 (Table 2), Burden, KBAC and SKAT-O recurrently fail

**Table 2. Power of six RVAS methods for 8 genes/architectures simulated using the 1000GP cohort and ClinVar and HGMD disease variants.** 100 Architectures were simulated for each gene. For BRCA1 and BRCA2 simulation was performed in missense and in LoF mode (see Material and Methods). Power is shown for VE = 1% and TIER levels 0.001 and 1e-04.

| Gene Method | BRCA1 MiSS | BRCA1 LoF | BRCA2 MiSS | BRCA2 LoF | BARD1 | BRIP1 | CHEK2 | PALB2 | |
|---|---|---|---|---|---|---|---|---|---|
| BURDEN | 94 | **100** | 0 | 0 | 2 | **100** | 24 | 78 | TIER = 0.001 |
| KBAC | 99 | **100** | 0 | 0 | 41 | **100** | 46 | 89 | |
| SKAT-O | 95 | **100** | 0 | 0 | 41 | **100** | 35 | 97 | |
| MiST | 0 | 0 | 0 | 0 | 0 | 0 | 0 | 0 | |
| HBMR | 79 | **100** | 97 | **100** | 37 | **100** | 83 | 16 | |
| BATI | 93 | **100** | 96 | **100** | 56 | **100** | 88 | 98 | |
| BURDEN | 77 | 95 | 0 | 0 | 0 | 84 | 1 | 22 | TIER = 1e-04 |
| KBAC | 89 | 99 | 0 | 0 | 5 | **100** | 10 | 35 | |
| SKAT-O | 85 | **100** | 0 | 0 | 0 | **100** | 1 | 49 | |
| MiST | 0 | 0 | 0 | 0 | 0 | 0 | 0 | 0 | |
| HBMR | 0 | 0 | 0 | 3 | 0 | 1 | 0 | 0 | |
| BATI | 78 | 99 | 84 | 93 | 15 | 90 | 50 | 82 | |

to call *BRCA2* (both missense and LoF versions), BARD1 and CHECK2 (Table 2). The performance of Burden, KBAC and SKAT-O varies considerably between genes, while MiST, HBMR and BATI show relatively small differences. Interestingly, the power plots at 0.5% VE look very similar when comparing Burden, KBAC and SKAT-O, indicating that these methods share the same strengths and weaknesses.

Only MiST and BATI are able to leverage categorical variant characteristics, here represented as functional annotations such as 'missense', 'LoF', 'indel'. As background LoF variants are rare we expected that both methods excel at predicting *BRCA1* and *BRCA2* under the LoF-architecture simulation. Indeed, for both methods we see a better performance for *BRCA1*-LoF and *BRCA2*-LoF compared to the *BRCA1*-missense and *BRCA2*-missense, respectively. For BATI, this difference is significant for *BRCA1* (p = 4.0e-13) and a tendency is found for *BRCA2* (p = 0.0025) for VE = 0.1 using Wilcoxon rank test. As a result, BATI predicts *BRCA2*-LoF at the highest significance level (TIER 0.01%), while all other methods perform poorly. *BRCA1*-LoF shows the highest ΔDIC value from all 8 risk genes, demonstrating that the BATI method strongly benefits from categorical functional annotations.

## Measuring the impact of categorical and numerical variant characteristics

BATI can account for individual variant characteristics under the assumption that similar variant-specific characteristics have a similar effect on the function of the protein and hence the phenotype. This is enabled by INLA, which provides Laplace approximation of the marginal posterior distributions for each of the elements of the parameter vector in Eq (4). Hence, we can obtain estimates of the parameters and their corresponding credible intervals in order to test if the disease risk is driven by a particular category of variants (e.g. LoF) or if one damage score discriminates pathogenic variants. Table 3 shows an example based on the analysis of one of our target gene architectures, BRCA1-LoF, where only LoF risk variants were added to a background of many non-pathogenic missense variants in the original samples. In this scenario we would expect that LoF variants have a significantly higher mean effect than the other variant categories, which is exactly what we observe (Table 3). For LoF SNV, the mean effect is significantly higher than 0 as shown by the 95% credible interval (non overlapping the 0 value), meaning that LoF SNVs show the strongest effect on disease predisposition for this gene. The impact of CADD on this scenario is weak, as all LoF SNVs have similar high CADD score.

**Table 3. BATI output of the genetic model estimates for the BRCA1-LoF gene architecture derived from one of the simulations.** Parameter estimates: mean effect (the mean increase of the variant effect on phenotype depending on the variant type, for numerical score CADD indicates the increase of the variant effect for a unit increase of the score), standard deviation of the mean (sd), 95% credible intervals of the mean(CI) and ΔDIC of the genetic model considered in the BATI-based RVAS test.

| | BRCA1-LoF architecture | | |
| --- | --- | --- | --- |
| | # variants | mean effect (sd) | 95%CI |
| Missense SNV | 26 | 0.291 (0.463) | (-0.618; 1.199) |
| LoF SNV | 17 | 0.629 (0.303) | (0.035; 1.225) |
| CADD | numerical | -0.003 (0.009) | (-0.024; 0.013) |
| *ΔDIC* | 14.946 | | |

## RVAS of chronic lymphocytic leukemia identifies candidate risk genes

Chronic lymphocytic leukemia (CLL) is a cancer of B-lymphocytes, which expands in the bone marrow, lymph nodes, spleen and blood. With the aim to identify the landscape of germline risk genes that can predispose an individual to CLL, we applied BATI and the other five competing RVAS methods integrated in rvGWAS. The CLL cohort of 436 cases was collected and sequenced following the guidelines of the International Cancer Genome Consortium (ICGC) [32] within the framework of the Spanish ICGC-CLL consortium[33]. In addition, 725 individuals from our Iberian cohort were used as controls. For the gene-wise RVAS test we preselected rare (MAF≤ 0.01 in our control cohort, ExAC and 1000GP) and potentially damaging variants (CADD score > 10). All RVAS methods were adjusted for the first 10 principal components to account for population stratification and technical biases. For BATI and MiST we additionally added the exonic function of the variants (i.e. LoF, missense, indel) and the CADD damage score as covariates. We tested all genes with a variant call rate of at least 95% and removed genes flagged by Allele Balance Bias (ABB)[34] as enriched with false positive variant calls (see S5 and S6 Tables and S8 Fig and S1 Text for details). BATI identified 12 candidates that passed the significance threshold of $10^{-4}$ (S7 Table). Among those, EHMT2 and COPS7A are promising CLL risk gene candidates. The heterodimeric methyltransferases EHMT1 and EHMT2 have recently been implicated with prognosis of CLL and CLL cell viability[35]. COPS7A (previous name COP9) is involved in the Transcription-Coupled Nucleotide Excision Repair (TC-NER) pathway and the COP9 signalosome complex (CSN) is involved in phosphorylation of p53/TP53, JUN, I-kappa-B-alpha/NFKBIA, ITPK1 and IRF8/ICSBP. However, replication of results in independent cohorts is required to evaluate these findings.

## Discussion

Here we presented a comprehensive framework, rvGWAS, to facilitate user-friendly and intuitive analysis of RVAS in case-control studies using whole genome or custom-captured next generation sequencing data. rvGWAS integrates data quality control and filtering, several existing rare variant association tests and the newly developed BATI test. We showed how BATI leverages both categorical and numerical variant characteristics and strongly benefits from their inclusion as covariates. We demonstrated BATI's significant gain in power if risk genes contain mostly LoF variants, while still performing at least as well as other methods when testing genes containing mostly missense variants.

Here we used CADD as a numerical functional impact score (representing deleteriousness). Other meta-predictors such as the more recently developed FatHMM[36], M-CAP[37] or REVEL[38] might improve results compared with CADD scores. With BATI, users can readily reanalyze cohorts with any functional impact or evolutionary conservation score of choice (or

multiple scores). The optimal selection of functional impact scores likely depends on the incidence of the disease or the targeted genomic regions. For instance, analysis of non-coding regions might benefit from specialized impact scores (e.g. FunSeq2), while the impact of very rare variants is better estimated by REVEL and M-CAP.

Model estimation when using complex data structures, including exome-wide genetic variants, numerical damage estimates and functional annotations, becomes computationally heavy. Therefore, existing tests resort to heuristics affecting accuracy (MiST) or are highly computationally intensive (HBMR). BATI addresses this issue by estimating the full model using Integrated Nested Laplace Approximation, which requires reasonable computational resources even when using complex data structures. INLA provides approximations to the posterior marginals of the latent variables, which are accurate and extremely fast to compute [18]. INLA was originally developed as a computationally efficient alternative to MCMC and presents two major advantages. On the one hand, INLA's fast speed allows it to work on models with huge dimensional latent fields and a large number of covariates at different hierarchical levels (for example in case of RVAS at the patient level and at the variant level). On the other hand, INLA treats latent Gaussian models in a unified way, thus allowing for greater automation of the inference process. Thanks to these characteristics, INLA has already been used in a great variety of applications[39–44]. While MiST only constructs a score test under the null hypothesis, BATI, leveraging the efficiency of INLA, can make inference based on full model estimation, and provides comprehensive information on estimates of model parameters. This leads to higher accuracy in terms of statistical inference and therefore higher power. Furthermore, BATI allows for the inclusion of many numerical or categorical features as covariates. Which other features, in addition to functional impact and functional annotation of variants, could be beneficial for association testing remains to be determined. Promising categories include variant call quality, tissue-specific gene expression measures, biological pathways or copy number variants.

Previous benchmark studies of RVAS tests typically relied on pure simulations of variants, for instance based on HapMap statistics, resulting in completely artificial cohorts[14]. Furthermore, simulations were often restricted to small regions of the genome, limiting their power for benchmarking exome-wide association tests. Simulated variant data is well-known to lack the complexity and noise-level of real data, resulting in overly optimistic benchmark performances and unrealistic expectations of the clinical researchers. Moreover, the use of random 'causal' variants hampers the benchmarking of methods that leverage characteristics of causal disease variants, which are enriched in high damage scores and high impact changes such as LoF variants. Here we combined real WES cohorts, representing realistic background noise, with real disease variants, featuring realistic functional impact profiles and variant distributions, to form semi-synthetic benchmark cohorts. We developed sampling methods allowing to test different disease architectures featuring various levels of variance explained in multiple risk genes. Furthermore, since the RVAS tests evaluated here use diverse criteria for statistical significance, e.g. p-value, Bayes factor or increase in deviance information criterion ($\Delta DIC$), we generated groups of cases and controls randomly from the 1000GP and Iberian cohorts without adding risk variants for the disease and benchmarked the six RVAS tests. As significant results from such tests can be considered false positives we could establish comparable significance thresholds across the different statistical criteria resulting in similar false positive rates. Hence, we estimated empirically the type I error rate per each RVAS test at different significant levels. In the case of $\Delta DIC$, a rule of thumb values of $\Delta DIC > 10$ are usually recommended to significantly distinguish different models under consideration, here the genetic model against a null model without genetic effects. This is in line with our results obtained from the type I error study in the 1000GP cohort to establish a significance level of 0.1%. In the

Iberian cohort, where the sample size is smaller, the specified threshold has been found to be slightly stricter ($>12$).

From our simulations, we show that methods vary substantially in power, especially for risk genes explaining a small fraction of the variance in a cohort. We found that differences between methods when VE is low (1% and 0.5%) are substantially more profound than previously appreciated, with some methods showing strongly fluctuating success rates for different genes and close to zero power at VE of 0.5%. For example, MiST showed favorable results on purely artificial benchmark sets[14], but performed poorly on our realistic WES cohorts, likely due to an issue with zero-inflated p-values caused by inappropriate handling of low variant counts. Specifically, MiST failed to identify any risk gene at low VE or low TIER thresholds. We further found that the performance patterns of Burden, KBAC and SKAT-O across the 8 risk gene architectures are highly similar when compared to MiST, HBMR and BATI. Burden, KBAC and SKAT-O fail to predict the same genes at 0.5% VE, namely BRCA2, BARD1 and CHEK2, which are characterized by high numbers of benign background variants. In those situations, limited sample sizes are a problem and it is therefore likely beneficial to combine Burden- and SKAT-type methods with completely different approaches to compensate for Burden and SKAT specific weaknesses.

The strong performance of BATI in terms of precision and recall comes at the price of longer run time (S8 Table). Inference based on full model estimation leads to a higher computational complexity and hence higher run time of BATI compared to all competing methods. The computational time and complexity of RVAS test methods is a concern, as exome and genome sequencing datasets have recently been increasing dramatically in sample size. However, the INLA implementation used by BATI (R-INLA project) facilitates the use of multiple cores, and scales close to linearly with the number of used cores, allowing for analysis of large cohorts on modern servers. Moreover, lowering the allele frequency threshold of included rare variants (e.g. from AF $< = 1\%$ to AF $< = 0.1\%$) for very large cohorts can dramatically reduce computation times. However, in order to facilitate BATI-based RVAS tests in a 'mega biobank scenario' with sample sizes larger than 100K, we recommend to perform a fast RVAS test, such as SKAT-O, to select the top-500 gene candidates and to re-analyze these with BATI to potentially achieve higher power.

In summary, leveraging variant characteristics and using the fast and accurate INLA model estimation, BATI outperforms existing RVAS test methods on realistic WES cohorts using real disease variants in 8 breast cancer risk genes, in hundreds of permutations. By facilitating integration of large numbers of covariates, BATI represents a flexible testing approach that can be further extended and enhanced in the future.

## Supporting information

**S1 Fig. Schematic overview of the rvGWAS platform.**
(TIF)

**S2 Fig. QC plots for 1,167 high quality samples from the Iberian cohort.** The 1,167 samples coming from Iberian population are used for benchmarking. rwGWAS QC showed the following QC statistics (A) Bar-plot for number of variants per sample, colored by variant type, (B) Barplot for number of variants per sample, colored by random assignment to cases (~1/3) or controls (~2/3), (C) Percentage of explained variance on first 9 PCA components, (D) Number of variants per gene in cases (x-Axis) and controls (y-axis). Each dot is one gene, while the red line shows the ratio of the number of cases and controls (1:2), (E) Histogram for number of mutations per sample, and (F) Projection on first 10 PCA components. Samples are colored by the center that performed the sequencing.
(TIF)

**S3 Fig. Variant Relative Risk distribution for MAF spectrum [0,0.01].**
(TIF)

**S4 Fig. Estimation of type 1 error rate (TIER) thresholds.** In 10 random splits of the 1000GP dataset into cases and controls, three commonly used significance levels thresholds (TIERs) are estimated: 0.05, 0.001 and 0.0001.
(TIF)

**S5 Fig.** P-value, Bayes Factor and ΔDIC distributions. For each RVAS test we created distribution of p-values/Bayes Factor/ Δ DIC with randomly assigned cases and controls in 1000GP dataset. In panels are shown distributions for (A) MiST, (B) MiST with p values larger than zero, (C) Burden, (D) KBAC, (E) SKAT-O, (F) HBMR, and (G) BATI RVAS test.
(TIF)

**S6 Fig. Power of RVAS methods for TIER levels 5%, 0.1% and 0.01% benchmarked in the Iberian cohort-based BRCA risk study.** Each dot in the plots represents one of 8 risk genes, and y-axis values show the fraction of 100 simulations in which the gene was called as significant. Variance explained of the incorporated risk variants (**A**) ~2%, (**B**) ~1%, and (**C**) ~0.5%.
(TIF)

**S7 Fig. Benchmark of statistical power to detect rare variant associations for 8 genes at 5% (red), 0.1% (green) and 0.01% (blue) TIERs in the Iberian cohort-based BRCA risk study.** Rare variants annotated for increased breast cancer risk were simulated into 1000GP dataset with cases and controls randomly assigned. Results per gene for 6 methods (Burden, KBAC, SKAT-O, MiST, HBMR and BATI) are shown for (**A**) 2%, (**B**) 1%, and (**C**) 0.5% variance explained between cases and healthy controls. Due to using real SNVs in the simulation the real variance explained per gene fluctuates slightly around the targeted VE (see S2 Fig). Red, blue and green lines indicate relaxed, medium and strict TIER thresholds, respectively.
(TIF)

**S8 Fig. Skewed allele balance for variants found in the gene FTCD.** The large number of variants found in FTCD in cases or controls that show a deviation from the expected 50:50 allele ratio expected for heterozygous SNVs, and the different distribution in cases and controls indicate a large number of false positive calls, leading to false gene-phenotype associations. This phenomenon is often caused by un-annotated segmental or tandem duplications in the reference genome, simple sequence repeats or copy gains in the samples. Using the method ABB we identified and excluded these genes from the RVAS test with the ICGC-CLL cohort.
(TIF)

**S1 Table. BRCA risk variants in ClinVar used for simulation as introduced causal variants.**
(DOCX)

**S2 Table. Number of variants in six BRCA risk genes in the 1000GP cohort before introduction of risk variants from ClinVar and HGMD (counting only rare coding or splicing variants with CADD $> 10$).**
(DOCX)

**S3 Table. Number of variants in six BRCA risk genes in the Iberian cohort before introduction of risk variants from ClinVar (counting only rare coding or splicing variants with CADD $> 10$).** Due to less than 100% variant call rates in some positions the number of possible cases and/or controls can be lower than total number of cases (389) and controls (778) used in the Iberian cohort.
(DOCX)

**S4 Table. P-value, Bayes Factor (HBMR) and ΔDIC (BATI) thresholds for Type I error rates (TIER) of 0.05, 0.001 and 1e-04 estimated on Iberian cohort.** We randomly permuted case and control labels 10 times and for each estimated empirical thresholds for each RVAS test. The median TIER values from 10 random permutations are used as thresholds for benchmark comparison.
(DOCX)

**S5 Table. Variant positions labelled as potential systematic errors by ABB.**
(DOCX)

**S6 Table. Potential false positive candidate genes.**
(DOCX)

**S7 Table. Significant risk genes for CLL detected by BATI, with support of up to 5 other RVAS tests.** Likely false positive gene associations labelled by ABB have been removed.
(DOCX)

**S8 Table. Run times of six RVAS methods (hours: minutes: seconds) using 12 cores for 898 cases and 912 controls randomly selected from the 1000GP cohort.** A total of 16676 genes were tested using an allele frequency threshold of AF$< = 0.01$ for all identified SNVs and indels.
(DOCX)

**S1 Text.** Provides additional details about a) the cohort simulation method, b) the integration of third-party rare variant association methods in rvGWAS, c) quality control analysis with rvGWAS, and d) evaluation of potential false positive rare variant calls detected within the chronic lymphocytic leukemia risk genes analysis.
(DOCX)

## Author Contributions

**Conceptualization:** Hana Susak, Raquel Rabionet, Geòrgia Escaramís, Stephan Ossowski.

**Data curation:** Hana Susak, German Demidov, Raquel Rabionet, Laura Domènech, Mattia Bosio, Francesc Muyas, Geòrgia Escaramís.

**Formal analysis:** Hana Susak, German Demidov.

**Funding acquisition:** Raquel Rabionet, Xavier Estivill, Stephan Ossowski.

**Investigation:** Hana Susak, Geòrgia Escaramís, Stephan Ossowski.

**Methodology:** Hana Susak, Laura Serra-Saurina, Geòrgia Escaramís, Stephan Ossowski.

**Project administration:** Xavier Estivill, Geòrgia Escaramís, Stephan Ossowski.

**Resources:** Xavier Estivill, Stephan Ossowski.

**Software:** Hana Susak, Geòrgia Escaramís.

**Supervision:** Geòrgia Escaramís, Stephan Ossowski.

**Validation:** German Demidov, Raquel Rabionet, Laura Domènech, Francesc Muyas.

**Visualization:** Hana Susak, German Demidov.

**Writing – original draft:** Hana Susak, Geòrgia Escaramís, Stephan Ossowski.

**Writing – review & editing:** Hana Susak, Laura Serra-Saurina, German Demidov, Raquel Rabionet, Laura Domènech, Mattia Bosio, Francesc Muyas, Xavier Estivill, Geòrgia Escaramís, Stephan Ossowski.

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
