## [Decision Letter · Decision Letter 0]

25 Jul 2020

Dear Dr. Escaramis,

Thank you very much for submitting your manuscript "Efficient and Flexible Integration of Variant Characteristics in Rare Variant Association Studies Using Integrated Nested Laplace Approximation" for consideration at PLOS Computational Biology.

As with all papers reviewed by the journal, your manuscript was reviewed by members of the editorial board and by several independent reviewers. In light of the reviews (below this email), we would like to invite the resubmission of a significantly-revised version that takes into account the reviewers' comments.

We cannot make any decision about publication until we have seen the revised manuscript and your response to the reviewers' comments. Your revised manuscript is also likely to be sent to reviewers for further evaluation.

Sincerely,

Yue Li, Ph.D.

Guest Editor

PLOS Computational Biology

Weixiong Zhang

Deputy Editor

PLOS Computational Biology

Reviewer's Responses to Questions

**Comments to the Authors:**

Reviewer #1: Review uploaded as attachment.

Reviewer #2: Susak et al. propose a novel rare variant burden test to leverage functional annotations of genetic variants. The INLA technique was applied to calculate the posterior. The deviance information criterion was used as a test statistic, and the false positive rate was controlled by permutation.

p.8 line 173-181: Is the model specification used in this study identical to MiST or not? I recommend the authors to clarify this. It's not clear to me whether the method was actually run with a full model that allowed for the dependency structure across variants. It sounds like they did, but I cannot find the details about this in the rest of the manuscript. If the authors ran the full dependency model, the authors need to present data on how good the estimated dependency structures are and whether the algorithm finds a meaningful dependency, in simulations and/or real data. If they are suggesting such a possibility for future studies, I would recommend clarifying this or moving it to discussions. I think the phrase that "this is enabled by INLA" is an over-statement without presenting the data.

Benchmark comparisons to MiST: It's great that the new method is more powerful than the previous methods. However, since BATI and MiST are based on similar model specification, it would be great to have some analyses done about why BATI works better with INLA. Is it because of the full dependency structure, what happens if you constrain this to be identical as MiST? Or is the higher accuracy because of calculating full DIC instead of a score test?

p.13 line 309-310: The prevalence of breast cancer is orders of magnitude higher than this. Based on the reference cited, I wonder if the authors perhaps confused incidence rates with prevalence. For the simulation of genetic architecture, the authors need the prevalence (because this is about the liability threshold), not the incidence rate. Simulated architecture with a prevalence of 0.00085 is interesting but unrealistically low not only for breast cancer but also for other typical common complex disorders. I don't think it's appropriate to simulate the allelic heterogeneity of typical common complex diseases under such an extreme prevalence.

Selection of causal variants: I think it's a great idea to use known ClinVar variants for simulation. However, there could be ascertainment biases in ClinVar variants, i.e. LoFs and missense variants with extreme CADD scores are more likely to be reported to ClinVar. And more subtle variants are less likely to be reported to ClinVar. It would be great if the authors run some of the simulated benchmarks with causal variants sampled from a mixture of both ClinVar and non-ClinVar variants.

**Have all data underlying the figures and results presented in the manuscript been provided?**

Reviewer #1: Yes

Reviewer #2: Yes

PLOS authors have the option to publish the peer review history of their article (what does this mean?). If published, this will include your full peer review and any attached files.

Reviewer #1: No

Reviewer #2: No
---

## [Decision Letter · Decision Letter 1]

4 Jan 2021

Dear Dr. Escaramis,

We are pleased to inform you that your manuscript 'Efficient and Flexible Integration of Variant Characteristics in Rare Variant Association Studies Using Integrated Nested Laplace Approximation' has been provisionally accepted for publication in PLOS Computational Biology.

Best regards,

Yue Li, Ph.D.

Guest Editor

PLOS Computational Biology

Weixiong Zhang

Deputy Editor

PLOS Computational Biology

Reviewer's Responses to Questions

**Comments to the Authors:**

Reviewer #1: The review has been uploaded as an attachment.

Reviewer #3: Well done. I have no further comments.

**Have all data underlying the figures and results presented in the manuscript been provided?**

Reviewer #1: Yes

Reviewer #3: None

PLOS authors have the option to publish the peer review history of their article (what does this mean?). If published, this will include your full peer review and any attached files.

Reviewer #1: No

Reviewer #3: No

---

## [Editor Report · Acceptance letter]

12 Feb 2021

PCOMPBIOL-D-20-00382R1 

Efficient and Flexible Integration of Variant Characteristics in Rare Variant Association Studies Using Integrated Nested Laplace Approximation

Dear Dr Escaramis,

I am pleased to inform you that your manuscript has been formally accepted for publication in PLOS Computational Biology. Your manuscript is now with our production department and you will be notified of the publication date in due course.

With kind regards,

Alice Ellingham
